# Generalizable Multi-Linear Attention Network

**Tao Jin**
Zhejiang University
jint_zju@zju.edu.cn

**Zhou Zhao**[*]
Zhejiang University
zhaozhou@zju.edu.cn

## Abstract

The majority of existing multimodal sequential learning methods focus on how to obtain powerful individual representations and neglect to effectively capture the multimodal joint representation. Bilinear attention network (BAN) is a commonly used integration method, which leverages tensor operations to associate the features of different modalities. However, BAN has a poor compatibility for more modalities, since the computational complexity of the attention map increases exponentially with the number of modalities. Based on this concern, we propose a new method called generalizable multi-linear attention network (MAN), which can associate more modalities in acceptable complexity with hierarchical approximation decomposition. Specifically, considering the fact that softmax attention kernels cannot be decomposed as linear operation directly, we adopt the addition random features mechanism to approximate the non-linear softmax functions with enough theoretical analysis. Furthermore, we also introduce the local sequential constraints, which can be combined with ARF conveniently, as positional information. We conduct extensive experiments on several datasets of corresponding tasks, the experimental results show that MAN could achieve competitive results compared with baseline methods, showcasing the effectiveness of our contributions.

## 1 Introduction

Multimodal learning draws increasing attention in recent years, which aims to process and understand the information from multiple modalities (i.e. vision, language, audio) with machine learning skills. Many endeavors have been devoted to the multimodal interaction and robust individual representation learning [29, 6, 39, 40]. Existing multimodal interaction methods could be categorized into Transformer-based [29, 6] and Non-Transformer-based methods [39, 40]. With deep stacked attention blocks [30] and suitable number of training samples, Transformer-based methods could achieve relatively good performances with relatively high computational complexity. After obtaining the individual representations of all the modalities, joint representation learning [5, 15, 14, 19, 36] is performed by leveraging simple matrix and vector operations. Bilinear functions have been widely utilized in this field. Specifically, bilinear pooling [15] employs low-rank decomposition to approximate high-order tensor operations for input vectors. Bilinear attention network (BAN) [14] uses a bilinear attention distribution, on top of low-rank bilinear pooling. In general, BAN exploits bilinear interactions between two groups of input channels and bilinear pooling extracts the joint representations for each pair of channels. Besides, bilinear pooling could be extended to multi-linear pooling with acceptable computational complexity. However, BAN has the limitation to generalize to more modalities due to the complexity. For example, when we want to integrate the features with the same space $\mathbb{R}^{d \times T}$ of $m$ modalities, we would obtain a high-order tensor $\mathbf{A} \in \mathbb{R}^{T^m}$ that captures multimodal interactions. It is observed that $\mathbf{A}$ scales exponentially with the number of modalities.

Motivated by the observations, in this paper, we propose a novel method called generalizable multi-linear attention network (MAN), which could be leveraged to integrate as many modalities as possible

---

[*]Corresponding Author

35th Conference on Neural Information Processing Systems (NeurIPS 2021).

with linear complexity. Concretely, we employ multi-linear pooling to extract the joint representations for multiple feature channels of different modalities. With the previous assumption, we could obtain $T^m$ joint representations for $m$ modalities in total. Following the main idea of BAN, an interaction tensor $\mathbf{A} \in \mathbb{R}^{T^m}$ represents the attention weights of the joint representations (in practise, we devise the combinational addition attention map). To reduce the computational complexity and space requirements, we devise a hierarchical approximation decomposition (HAD) mechanism for the high-order tensor $\mathbf{A}$, leading to the efficient approximation decomposition. Specifically, considering the fact that softmax attention kernels cannot be decomposed as linear operation directly, we devise the addition random features (ARF) mechanism to approximate the non-linear softmax function. Furthermore, we facilitate the sparse attention distributions by leveraging local sequential constraints (LSC), which can be combined with ARF conveniently. We conduct extensive experiments on three tasks, the experimental results show that MAN could achieve competitive results compared with the state-of-the-art methods. To sum up, the contributions of our work are four-fold:

- We propose the multi-linear attention network (MAN), which is the extension of bilinear attention network (BAN), to learn and utilize multi-linear attention distributions on top of the multi-linear pooling operations. To reduce the computational complexity and space requirements, we devise a hierarchical approximation decomposition mechanism for the combinational addition interaction tensor, leading to the efficient approximation decomposition.

- Most importantly, considering that softmax attention kernels cannot be decomposed as linear operation directly, we adopt the addition random features (ARF) approach to approximate the non-linear softmax functions with enough theoretical analysis.

- To process multimodal sequential features, we introduce the local sequential constraints (LSC), which can be combined with ARF conveniently.

- To be compatible with Transformer, the multi-head and residual properties are introduced into MAN. We conduct extensive experiments on three different tasks, the experimental results show that MAN could achieve competitive results compared with the baseline methods.

## 2 Approach

In this section, we first introduce the low-rank multi-linear pooling, which is the extension of bilinear pooling [15]. Then we illustrate the details of multi-linear attention network.

### 2.1 Multi-Linear Pooling

We first review the low-rank bilinear pooling. [15] proposes to reduce the rank of bilinear weight matrix $\mathbf{W}_i \in \mathbb{R}^{d_1 \times d_2}$ with CP decomposition [7]. $\mathbf{W}_i$ could be rewrited as the multiplication of two smaller matrices $\mathbf{U}_i \mathbf{V}_i^T$, where $\mathbf{U}_i \in \mathbb{R}^{d_1 \times d}$ and $\mathbf{V}_i \in \mathbb{R}^{d_2 \times d}$, $d$ denotes the value of rank with the constraint $d \leq \min(d_1, d_2)$. We calculate the scalar output $f_i$ as follows:

$$f_i = \mathbf{x}^T \mathbf{W}_i \mathbf{y} \approx \mathbf{x}^T \mathbf{U}_i \mathbf{V}_i^T \mathbf{y} = \mathbf{1}^T \left( \mathbf{U}_i^T \mathbf{x} \odot \mathbf{V}_i^T \mathbf{y} \right) \tag{1}$$

where $\mathbf{1} \in \mathbb{R}^d$ consists of ones and $\odot$ denotes element-wise multiplication, note that we omit the bias term for convenience. When the output is a vector $\mathbf{f} \in \mathbb{R}^c$, the $\mathbf{1}$ is replaced by a pooling matrix:

$$\mathbf{f} = \mathbf{P}^T \left( \mathbf{U}^T \mathbf{x} \odot \mathbf{V}^T \mathbf{y} \right) \tag{2}$$

where $\mathbf{P} \in \mathbb{R}^{d \times c}$, $\mathbf{U} \in \mathbb{R}^{d_1 \times d}$, and $\mathbf{V} \in \mathbb{R}^{d_2 \times d}$. By leveraging the pooling matrix $\mathbf{P}$, the number of parameters is reduced significantly. Based on the bilinear pooling, we introduce the calculation process of multi-linear pooling. Suppose the number of modalities is $m$ and the feature (channel) dimensions of them are $\{d_j | j \in [1, m]\}$, Eq. 2 is extended to:

$$\mathbf{f} = \mathbf{P}^T (\prod_{j=1}^m \mathbf{U}_j^T \mathbf{v}_j) = \mathbf{P}^T (\mathbf{U}_1^T \mathbf{v}_1 \odot \mathbf{U}_2^T \mathbf{v}_2 \odot ... \odot \mathbf{U}_m^T \mathbf{v}_m) \tag{3}$$

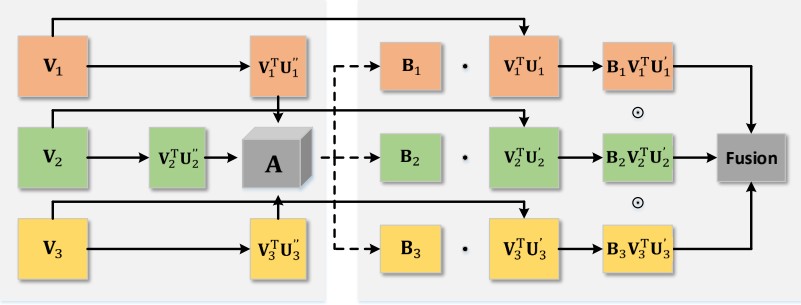

Figure 1: The core module of multi-linear attention network (only one head), where we provide detailed input and output of each step, making it easy to understand. Note that, for convinience, we give an example of three modalities.

where $\mathbf{v}_j \in \mathbb{R}^{d_j}$ and $\mathbf{U}_j \in \mathbb{R}^{d_j \times d}$ denote the feature vector of $j$-th modality and corresponding projection matrix. Theoretically, $\mathbf{P}$ and $\{\mathbf{U}_j | j \in [1, m]\}$ are the components of $d$-rank approximation of the high-order tensor.

## 2.2 Multi-Linear Attention Network

Since the features of most modalities are multi-channel (for example, video features have temporal dimension), we propose MAN to learn and utilize the multi-linear attention distributions on top of the multi-linear pooling operations. Suppose the feature matrices of $m$ modalities are $\{\mathbf{V}_j \in \mathbb{R}^{d_j \times T_j} | j \in [1, m]\}$, we could obtain $\prod_{j=1}^{m} T_j$ groups of joint representations. To combine these information, we introduce the multi-linear attention map $\mathbf{A} \in \mathbb{R}^{\prod_{j=1}^{m} T_j}$ as follows:

$$f_k^{'} = ((\mathbf{A} \times_1 (\mathbf{V}_1^T \mathbf{U}_1^{'})_k) \times_2 (\mathbf{V}_2^T \mathbf{U}_2^{'})_k) \times_3 ... \times_m (\mathbf{V}_m^T \mathbf{U}_m^{'})_k \tag{4}$$

where $\mathbf{U}_j^{'} \in \mathbb{R}^{d_j \times K}$, $(\mathbf{V}_j^T \mathbf{U}_j^{'})_k \in \mathbb{R}^{T_j}$, the operator $\times_j$ denotes the $j$-th mode product between a tensor and a vector, and $f_k^{'}$ denotes the $k$-th element of intermediate representation. Note that Eq. 4 is a multi-linear model for $m$ groups of input channels, where $\mathbf{A}$ is a multi-linear weight matrix. To make it easier to understand, Eq. 4 could be rewritten as follows:

$$f_k^{'} = \sum_{t_1=1}^{T_1} ... \sum_{t_m=1}^{T_m} \mathbf{A}_{t_1, t_2, ..., t_m} (\prod_{j=1}^{m} \mathbf{V}_{j, t_j}^T \mathbf{U}_{j, k}^{'}) \tag{5}$$

where $\mathbf{V}_{j, t_j} \in \mathbb{R}^{d_j}$ denotes the $t_j$-th channel of input $\mathbf{V}_j$ and $\mathbf{U}_{j, k}^{'} \in \mathbb{R}^{d_j}$ denotes the $k$-th column of $\mathbf{U}_j$, $\mathbf{A}_{t_1, t_2, ..., t_m}$ denotes an element in tensor $\mathbf{A}$ with indices $[t_1, t_2, ..., t_m]$. The intermediate result $\mathbf{f}^{'} \in \mathbb{R}^K$ consists of $K$ elements $[f_1^{'}, ..., f_k^{'}, ..., f_K^{'}]$. Then the multi-linear joint representation is $\mathbf{f} = \mathbf{P}^T \mathbf{f}^{'}$ where $\mathbf{f} \in \mathbb{R}^c$ and $\mathbf{P} \in \mathbb{R}^{K \times c}$. For convenience, we define the multi-linear attention network as a function of $m$ multi-channel inputs parameterized by a multi-linear attention map as follows:

$$\mathbf{f} = \text{MAN}(\{\mathbf{V}_j | j \in [1, m]\}; \mathbf{A}) \tag{6}$$

**Combinational Additive Attention Map.** We argue that multi-linear attention map should have **several properties**: **(1)** The attention logits are calculated by leveraging the interaction information between different modalities. Each modality is in a symmetric (equal) position. **(2)** Following the main idea of BAN, the attention map can be calculated with Hadamard product and tensor operations similarly. However, since the absolute values of most multiplicative factors are between 0 and 1, directly employing softmax function may result in even attention distributions when the attention map is large. **(3)** As for the processing of multimodal sequential features, the time interval (relative position) of different modalities should be considered. In general, locally close moments are more

relevant. Following the idea of Transformer, most positional encoding methods can be integrated with the inner product of two vectors. Therefore, we utilize simply combinational addition operation (CAO) to replace the Hadamard product. The attention map $\mathbf{A}$ is defined as follows:

$$\mathbf{A} := \text{softmax}(\mathcal{A}), \quad \mathcal{A}_{t_1, t_2, \ldots, t_m} = (\sum_{j=1}^{m} \sum_{q=1, \neq j}^{m} (\mathbf{V}_{q,t_q}^{T} \mathbf{U}_q^{''})(\mathbf{U}_j^{''T} \mathbf{V}_{j,t_j})) \tag{7}$$

where $\mathbf{U}_j^{''} \in \mathbb{R}^{d_j \times K^{'}}$, $\mathcal{A} \in \mathbb{R}^{\prod_{j=1}^{m} T_j}$ and the softmax function is applied element-wisely along all the dimensions. In Eq. 7, the number of additive items $(\mathbf{V}_{q,t_q}^{T} \mathbf{U}_q^{''})(\mathbf{U}_j^{''T} \mathbf{V}_{j,t_j})$ is positively correlated with the number of modalities, resulting in the bigger difference between elements and sparser attention distributions. Besides, the combinational addition operation has satisfactory compatibility with local sequential constraints introduced in the subsequent section.

**Local Sequential Constraints (LSC) for Multimodal Temporal Features.** When the number of modalities increases, the attention tensor $\mathbf{A}$ becomes larger and the probability distributions are much denser. As mentioned in the previous section, the features of temporal proximity are more relevant. So we devise an attenuation function related to the time steps of associated modalities. Suppose the number of time steps is $T$, we assign each time step a $\frac{T}{s}$-dimensional vector ($\frac{T}{s}$ denotes the number of chunks and $s$ adjusts the scale), which consists of $e$ and $-e$. For example, the temporal vectors for the 1-st and 2-nd temporal chunks are $[e, -e, -e, \ldots, -e]$ and $[e, e, -e, \ldots, -e]$, the inner product result is $(\frac{T}{s} - 1) \times e^2$. Similarly, the inner product result between any $t_1$ and $t_2$ time steps are $(\frac{T}{s} - |t_2 - t_1|) \times e^2$, which fits the property of attenuation function. In practice, we simply rewrite the Eq. 7, concatenating the temporal vectors to the features of different modalities:

$$\mathcal{A}_{t_1, t_2, \ldots, t_m} = (\sum_{j=1}^{m} \sum_{q=1, \neq j}^{m} \left[ \mathbf{V}_{q,t_q}^{T} \mathbf{U}_q^{''}, \mathbf{E}_{t_q} \right] \left[ \mathbf{V}_{j,t_j}^{T} \mathbf{U}_j^{''}, \mathbf{E}_{t_j} \right]^{T}) \tag{8}$$

To balance the weights of modality association and local constraints, we adjust the value of $e$.

**Hierarchical Approximation Decomposition.** Considering the fact that the size of combinational additive interaction tensor $\mathbf{A}$ scales exponentially to the number of modalities. We devise a hierarchical decomposition mechanism to cope with this problem. Without restrictions, $\mathbf{A}$ can be decomposed with CP algorithm:

$$\mathbf{A} = \bigotimes_{j=1}^{m} \mathbf{B}_j, \quad \mathbf{A}_{t_1, t_2, \ldots, t_m} = \mathbf{1}^{T}(\prod_{j=1}^{m} \mathbf{B}_{j,t_j}) \tag{9}$$

where $\mathbf{1} \in \mathbb{R}^{H}$, $\mathbf{B}_j \in \mathbb{R}^{H \times T_j}$ and $H$ denotes the rank value, $\otimes$ is the defined outer product operation for multiple matrices, which is a bit different from traditional vectorial operation. The right equation is the detailed calculation for each element, where $\mathbf{B}_{j,t_j} \in \mathbb{R}^{H}$ denotes the $t_j$-th channel of $\mathbf{B}_j$. We directly substitude the Eq. 8 into Eq. 5 as follows[2]:

$$\mathbf{f}^{'} = \sum_{t_1=1}^{T_1} \ldots \sum_{t_m=1}^{T_m} (\mathbf{1}^{T}(\prod_{j=1}^{m} \mathbf{B}_{j,t_j}))(\prod_{j=1}^{m} \mathbf{V}_{j,t_j}^{T} \mathbf{U}_j^{'}) = \mathbf{1}^{T}(\prod_{j=1}^{m} \mathbf{B}_j \mathbf{V}_j^{T} \mathbf{U}_j^{'}) \tag{10}$$

where we obtain a simple conclusion. $\mathbf{B}_j \in \mathbb{R}^{H \times T_j}$ and $\mathbf{U}_j^{'} \in \mathbb{R}^{d_j \times K}$ can be treated as two linear projection weights for different dimensions, respectively.

**Addition Random Features.** Although Eq. 9 is quite simple and pretty, the high-order tensor $\mathbf{A}$ cannot be decomposed directly due to the non-linearity caused by softmax function. Thus, it is necessary to find a substitute approximate solution. Inspired by the Performer [3], we propose the addition random features mechanism, which can be treated as an extension of vanilla random features mechanism for more than two modalities. The detailed process is shown as follows[1]:

---

[2]The detailed proof and complete algorithm process are shown in the appendix.

$$\text{SM}(\{\mathbf{v}_j | j \in [1,m]\}) = \mathbb{E}_{\boldsymbol{w} \sim \mathcal{N}(0, \mathbf{I}_{K'})} \left[ \prod_{j=1}^{m} \exp\left( \boldsymbol{w}^\top \mathbf{v}_j - \frac{\|\mathbf{v}_j\|^2}{2} \right) \right] \tag{11}$$

where $v_j$ denotes the feature vector of $j$-th modality, $\text{SM}(\{\mathbf{v}_j | j \in [1,m]\})$ is equal to $\exp(\sum_{j=1}^{m} \sum_{k=1,\neq j}^{m} \mathbf{v}_k^T \mathbf{v}_j)$, $\mathbb{E}$ and $\mathcal{N}(0, \mathbf{I}_{K'})$ denote expectation and sampling distribution. The detailed derivations can be found in the supplementary materials. Eq. 10 is employed to calculate $\mathbf{A}_{t_1, t_2, \dots, t_m}$ directly. $\mathbf{B}_j$ in Eq. 8 can be obtained by project the features of $j$-th modality directly with ARF mechanism. In this way, the number of random features is equal to the rank value $H$ of $\mathbf{A}$. Note that softmax function consists of exponential operation and sum normalization, we need the value of normalized denominator. In practice, we directly employ $\{\mathbf{B}_j | j = [1,m]\}$ to calculate it[1]:

$$\sum \mathbf{A} = \sum \left( \prod_{j=1}^{m} \mathbf{B}_j \cdot \mathbf{1} \right) \tag{12}$$

where $\sum$ denotes the sum of all elements in the tensor and $\mathbf{1} \in \mathbb{R}^{T_j}$. We could keep $\sum \mathbf{A}$ and utilize the integration result $\mathbf{f}'$ to divide it. Note that, during the calculation of ARF for softmax function, a reduction factor $\frac{1}{H}$ is introduced into both numerator and denominator, thus it directly cancel out. We could treat the expectation operation in Eq. 10 as the summation operation. The core module of MAN is shown in Fig. 1.

**Multi-head Multi-linear Attention.** Following the main idea of Transformer [30] that projecting the features into multiple representation spaces could capture more information, we also adopt the multi-head strategy in the multi-linear attention. In practice, we add multiple linear mappings to the input features before fusing them (i.e. multiple $\mathbf{U}_j'$ and $\mathbf{U}_j''$ for $\{\mathbf{V}_j^T \mathbf{U}_j', \mathbf{V}_j^T \mathbf{U}_j'' | j \in [1,m]\}$ ). The results of all the heads are concatenated. The calculation follows the process described above.

**Stacked Multi-linear Attention.** The residual connection is proved to be effective in most structures [8]. Therefore, such strategy can also be introduced into multi-linear attention network. Specifically, Eq. 6 is modified as:

$$\mathbf{f}_{n+1}^l = \text{MAN}(\mathbf{f}_n^l, \{\mathbf{V}_j | j \in [1,m], j \neq l\}; \mathbf{A}) \cdot \mathbf{1}^T + \mathbf{f}_n^l \tag{13}$$

where $\mathbf{1} \in \mathbb{R}^{T_l}$, $\mathbf{f}_n^l \in \mathbb{R}^{d_l \times T_l}$ denotes the $n$-th output of $l$-th modality and $\mathbf{f}_0^l = \mathbf{V}_l$ (if $\frac{d_1}{g} = \frac{d_2}{g} = \dots = \frac{d_m}{g} = K = c$, $g$ denotes the number of heads).

**Complexity.** Without approximation decomposition, the computational complexity increases exponentially with respect to the number of modalities. Concretely, we need $\prod_{j=1}^{m} T_j$ low-rank multimodal pooling operations, leading to the unfriendly complexity $O(\prod_{j=1}^{m} T_j)$. While the complexity scales linearly $O(\sum_{j=1}^{m} T_j)$ with the number of modalities by leveraging approximation decomposition.

## 3 Experiments

### 3.1 Datasets

We evaluate MAN on three challenging tasks, multimodal sentiment analysis, multimodal speaker traits recognition, and multimodal video retrieval. In this section, we provide a brief introduction of the datasets (CMU-MOSI [41] for multimodal sentiment analysis, POM [25] for multimodal speaker traits recognition, MSR-VTT [33] and LSMDC [28] for multimodal video retrieval).

**CMU-MOSI:** The CMU-MOSI dataset is a collection of 93 opinion videos from YouTube movie reviews. Each video consists of multiple opinion segments (2199 segments in total) and each segment is annotated with the score in the range $[-3, 3]$, where $-3$ and $3$ indicate highly negative and positive. There are 1284 segments in the training set, 229 in the validation set, and 686 in the test set.

**POM:** POM is a multimodal speaker traits recognition dataset made up of 903 movie review videos. Each video is annotated for various personality and speaker traits, specifically: Confident (con),

| Model \Metric | BA | F1 | MAE | Corr | MA |
|---|---|---|---|---|---|
| MV-LSTM [27] | 73.9 | 74.0 | 1.019 | 0.601 | 33.2 |
| TFN [38] | 73.9 | 73.4 | 1.040 | 0.633 | 32.1 |
| MARN [40] | 77.1 | 77.0 | 0.968 | 0.625 | 34.7 |
| MFN [39] | 77.4 | 77.3 | 0.965 | 0.632 | 34.1 |
| RAVEN [31] | 78.0 | – | 0.915 | 0.691 | – |
| LMF [19] | 76.4 | 75.7 | 0.912 | 0.668 | 32.8 |
| MAN (Non-Transformer) | **79.4** | **79.3** | **0.894** | **0.697** | **37.8** |
| MulT [29] | **83.0** | 82.8 | 0.870 | 0.698 | 40.0 |
| MAN (Transformer) | 82.7 | **83.0** | **0.866** | **0.707** | **40.5** |

Table 1: MAN achieves superior performances over baseline models for CMU-MOSI dataset. We report BA (binary accuracy), F1, Corr (Pearson Correlation Coefficient), and MA (Multi-class accuracy, all higher is better), MAE (Mean-absolute Error, lower is better).

| Model \ Trait | Con | Pas | Voi | Dom | Cre | Viv | Exp | Ent |
|---|---|---|---|---|---|---|---|---|
| | MA7 | MA7 | MA7 | MA7 | MA7 | MA7 | MA7 | MA7 |
| MV-LSTM [27] | 25.6 | 28.6 | 28.1 | 34.5 | 25.6 | 32.5 | 32.5 | 29.6 |
| TFN [38] | 24.1 | 31.0 | 31.5 | 34.5 | 24.6 | 25.6 | 27.6 | 29.1 |
| MARN [40] | 29.1 | 33.0 | - | - | 31.5 | - | - | - |
| MFN [39] | 34.5 | 35.5 | 37.4 | 41.9 | 34.5 | 36.9 | 36.0 | 37.9 |
| LMF [29] | 35.9 | 35.9 | 34.8 | 39.6 | 34.5 | 35.9 | 37.8 | 36.5 |
| MAN (Non-Transformer) | **37.5** | **38.3** | **37.9** | **43.6** | **37.0** | **38.7** | **40.5** | **38.9** |
| MulT [29] | 34.5 | 34.5 | 36.5 | 38.9 | 37.4 | 36.9 | 37.9 | 39.4 |
| MAN (Tranformer) | **37.9** | **38.9** | **37.9** | **44.8** | **37.9** | **39.9** | **40.9** | **40.4** |
| Model \ Trait | Res | Tru | Rel | Out | Tho | Ner | Per | Hum |
| | MA5 | MA5 | MA5 | MA5 | MA5 | MA5 | MA7 | MA5 |
| MV-LSTM [27] | 33.0 | 52.2 | 50.7 | 38.4 | 37.9 | 42.4 | 26.1 | 38.9 |
| TFN [38] | 30.5 | 38.9 | 35.5 | 37.4 | 33.0 | 42.4 | 27.6 | 33.0 |
| MARN [40] | 36.9 | - | 52.2 | - | - | 47.3 | 31.0 | 44.8 |
| MFN [39] | 38.4 | 57.1 | 53.2 | 46.8 | **47.3** | **47.8** | 34.0 | 47.3 |
| LMF [29] | 35.5 | 54.2 | 53.2 | 44.8 | 42.7 | 43.5 | 34.9 | 45.8 |
| MAN (Non-Transformer) | **39.6** | **59.5** | **55.2** | **47.4** | 46.5 | 47.0 | **37.2** | **49.8** |
| MulT [29] | 41.4 | 60.6 | 54.2 | 43.3 | **49.3** | 46.3 | 33.5 | 43.3 |
| MAN (Transformer) | **41.9** | **61.4** | **55.2** | **49.1** | 44.8 | **48.3** | **38.3** | **51.2** |

Table 2: MAN achieves superior performances over baseline models in POM dataset (multimodal personality traits recognition). MA(5,7) denotes multi-class accuracy for (5,7) classes.

Passionate (Pas), Voice Pleasant (Voi), Dominant (Dom), Credible (Cre), Vivid (Viv), Expertise (Exp), Entertaining (Ent), Reserved (Res), Trusting (Tru), Relaxed (Rel), Outgoing (Out), Thorough (Tho), Nervous (Ner), Persuasive (Per) and Humorous (Hum). The training, validation, and test set distributions are approximately 600, 100, and 203, respectively.

**MSR-VTT:** MSR-VTT is composed of 10K YouTube videos, collected using 257 queries from a commercial video search engine. Each video is 10 to 30s long, and is paired with 20 natural sentences describing it, obtained from Amazon Mechanical Turk Workers. We report results on the train/test splits introduced in [34] that uses 9000 videos for training and 1000 for testing.

**LSMDC:** It contains 118081 short video clips (about 45s) extracted from 202 movies. Each clip is annotated with a caption, extracted from either the movie script or the audio description. The test set is composed of 1000 videos, from movies not presented in the training set.

| Model \Metric | BA | F1 | MAE | Corr | MA |
|---|---|---|---|---|---|
| w/o CAO | 78.4 | 78.3 | 0.924 | 0.668 | 35.2 |
| w/o LSC | 78.4 | 78.3 | 0.928 | 0.673 | 36.5 |
| w/o HAD | 79.4 | 79.3 | 0.901 | 0.693 | 38.1 |
| MAN (Non-Transformer) | 79.4 | 79.3 | 0.894 | 0.697 | 37.8 |
| w/o CAO | 81.5 | 81.6 | 0.880 | 0.692 | 38.9 |
| w/o LSC | 81.4 | 81.3 | 0.883 | 0.698 | 39.8 |
| w/o HAD | 82.6 | 82.8 | 0.872 | 0.707 | 40.5 |
| MAN (Transformer) | 82.7 | 83.0 | 0.866 | 0.707 | 40.5 |

Table 3: Ablation study on CMU-MOSI dataset.

| Metric \Model | w/o CAO | w/o HAD | w/o LSC | MAN |
|---|---|---|---|---|
| FLOPs | $8.31 \times 10^6$ | $10.23 \times 10^6$ | $3.07 \times 10^5$ | $3.87 \times 10^5$ |

Table 4: Computational complexity of different variants on CMU-MOSI, note that we only consider the FLOPs of one-layer integration module.

## 3.2 Experiments for Multimodal Sentiment Analysis and Speaker Traits Recognition

**Data Preprocessing:** Each dataset (CMU-MOSI, POM) consists of three modalities, including textual, visual, and audio modalities. For textual features, we employ the pre-trained 300-dimensional Glove embeddings [24]. For visual features, we utilize Facet [11] to indicate 35 facial action units, which records facial muscle movement for representing the basic and advanced emotions. For audio features, we use COVAREP [4] acoustic analysis framework. To align the different modalities along the temporal dimension, we perform word alignment with P2FA [37], which aligns the modalities at the word granularity.

**Experimental Details:** Transformer-based and Non-Transformer-based methods both have their practical meanings of existence. For example, without enough labeled data, non-transformer-based methods may be an optimal solution due to less computational complexity. Henceforce, we combine MAN with both Transformer-based and Non-Transformer-based backbones and make corresponding comparisons with existing methods. We refer to [29] and [19] for our implementation of backbones, respectively. Note that we do not remove the original structures of [29] and [19], but add MAN as an extra module. We fuse their results for the tasks. The hyperparameters of MAN include Adam learning rate 0.001, the structure of integration network (1 layer of integration block, with hidden sizes of 40, number of heads 10, number of random features 24). "Hidden size" denotes the common size of $d_1, d_2,...,d_m$. We divide all the time steps into 4 chunks and apply local sequential constraints. Furthermore, for the specific hyperparameters of backbones, we use similar values as original papers.

**Results:** Table 1 presents the overall comparison of MAN and existing methods on CMU-MOSI dataset. As for the Non-Transformer-based methods, we could observe that MV-LSTM, MFN, MARN, RAVEN perform worse than MAN as they pay more attention to multimodal interaction and representation learning, and ignore the importance of subsequent multimodal integration. Besides, MAN achieves the best performances on all the metrics among the existing multimodal integration methods TFN, LMF and gains a large margin. Since TFN, LMF neglect the fine-grained temporal interaction which includes rich structured information for multimodal modeling. Particularly, the performance of MAN decreases the MAE from 0.912 to 0.894 compared to the best counterparts. As for the Transformer-based methods, MAN also performs better than MulT on all the metrics, which demonstrates that, to some extent, subsequent multimodal joint representation learning could be integrated with the early multimodal interaction, even though only multimodal interaction has been able to achieve competitive performances. In general, the best performances of MAN attribute to the advanced hierarchical approximation decomposition which imposes fine-grained temporal correlation with acceptable computational complexity, as well as the addition random features mechanism that provides support for the approximation decomposition. Table 2 shows the experimental results of different methods on speaker traits recognition dataset POM, where we report the multi-class accuracy of all the traits. The similar observation could be found from the table, MAN achieves competitive performances compared with both Transformer-based and Non-Transformer-based methods on most of the traits. The only difference is that the metrics of Transformer-based and Non-Transformer-based

methods are close. Therefore, even the Transformer has great potential in multimodal learning, its ability may be limited by the number of training samples. We argue that Non-Transformer-based methods still have their meanings in the real-world applications.

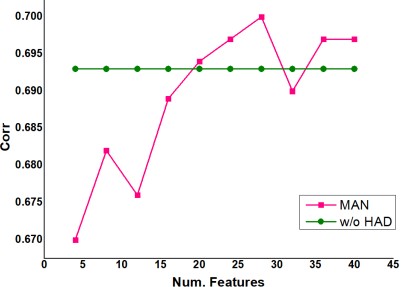 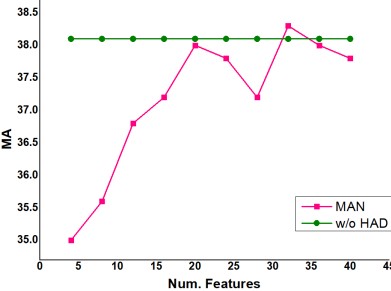

Figure 2: The evaluation of different numbers of random features, where we report the values of Corr and MA on CMU-MOSI.

**Ablation Study:** We set some control experiments on CMU-MOSI to verify the effectiveness of MAN and the results are shown in Table 3 and Table 4, where "w/o CAO" denotes the model without combinational addition operation in Eq. 7, we simply employ the continued Hadamard product and do not decompose the attention map, in this way, the local constraints also cannot be utilized. "w/o HAD" denotes the model without hierarchical approximation decomposition, but with other contributions. "w/o LSC" denotes the model with all the contributions except for local sequential constraints. We could observe that "w/o HAD" performs similarly to MAN with much more computational complexity, since "w/o HAD" needs to generate a high-order attention map with exponential complexity. Besides, MAN and "w/o HAD" perform much better than "w/o CAO", since "w/o CAO" generates denser attention distributions and cannot capture really important information. Additionally, by comparing the metrics of "w/o LSC" and MAN, we argue that the local constraints have impacts on better capturing the correlation between multimodal sequential features. Since we utilize addition random features mechanism to approximate the softmax function, the number of random features should be considered. We examine the performances of MAN (Non-Transformer) on CMU-MOSI with different values of $H$. As shown in Fig. 2, when the value of $H$ is big enough, MAN achieves competitive performances of Corr and MA close to "w/o HAD".

### 3.3 Experiments for Multimodal Video Retrieval

**Data Preprocessing:** The videos contain abundant multimodal information. Thus, we use multiple pre-trained models for extracting features. Concretely, we utilize following seven experts: **Motion** embeddings are extracted from S3D [32] trained on the kinetics dataset. **Scene** embeddings are extracted with DenseNet-161 [10] trained on the Places365 dataset [42]. **OCR** embeddings are extracted in three stages. First, the pixel link text detection model is used to detect the overlaid text. Then, the detected boxes are passed through the text recognition model. Finally, each character sequence is encoded using a word2vec embedding. **Audio** embeddings are obtained with a VGGish model, trained on the YouTube-8m dataset. **Speech** features are extracted using the Google Cloud speech API, to extract word tokens from the audio stream, which are then encoded via pre-trained word2vec embeddings [22]. **Face** features are extracted by ResNet-50 [8] trained for face classification on the VGGFace2 dataset. **Appearance** features are extracted from the final global average pooling layer of SENet-154 [9] trained on ImageNet.

**Experimental Details:** We also combine MAN with Transformer-based and Non-Transformer-based backbones. For MAN (Non-Transformer), each word of the text query is encoded with pre-trained word2vec embeddings and then passed through the GPT model. For MAN (Transformer), we adopt the basic structure of [6] and train the word vectors from scratch. We refer to [18] and [6] for the implementation of backbones and keep the original structures, only adding MAN as an extra module. We further fuse their results. The hyperparameters of MAN include Adam learning rate $5 \times 10^{-5}$, which we decay by a multiplicative factor $0.95$ every $1000$ optimization steps, the structure of integration network (1 layer of integration block, with hidden size $512$, number of heads $8$, number of random features $512$). "Hidden size" denotes the common size of $d_1, d_2,...,d_m$. We divide the time steps into 10 chunks and employ local sequential constraints. We also use the same hyperparameters of specific backbones as the original papers.

| Model \Metric | Text $\longrightarrow$ Video | | | | Video $\longrightarrow$ Text | | | |
|---|---|---|---|---|---|---|---|---|
| | R@1↑ | R@5↑ | R@10↑ | MdR↓ | R@1↑ | R@5↑ | R@10↑ | MdR↓ |
| JSFusion [34] | 10.2 | 31.2 | 43.2 | 13 | - | - | - | - |
| HT [21] | 14.9 | 40.2 | 52.8 | 9 | - | - | - | - |
| CE [18] | 20.9 | 48.8 | 62.4 | 6 | 20.6 | 50.3 | 64.0 | 5.3 |
| MAN (Non-Transformer) | **21.4** | **51.0** | **63.5** | **5** | **21.6** | **51.3** | **65.4** | **5** |
| MMT [6] | **24.6** | 54.0 | 67.1 | **4** | **24.4** | 56.0 | 67.8 | **4** |
| MAN (Transformer) | 24.1 | **55.7** | **68.1** | **4** | **24.4** | **57.1** | **68.5** | **4** |

Table 5: Retrieval performances on the MSR-VTT dataset.

| Method \Metric | Text $\longrightarrow$ Video | | | | Video $\longrightarrow$ Text | | | |
|---|---|---|---|---|---|---|---|---|
| | R@1↑ | R@5↑ | R@10↑ | MdR↓ | R@1↑ | R@5↑ | R@10↑ | MdR↓ |
| CT-SAN [35] | 5.1 | 16.3 | 25.2 | 46 | - | - | - | - |
| JSFusion [34] | 9.1 | 21.2 | 34.1 | 36 | - | - | - | - |
| CCA [16] (rep. by [20]) | 7.5 | 21.7 | 31.0 | 33 | - | - | - | - |
| MEE [20] | 10.1 | 25.6 | 34.6 | 27 | - | - | - | - |
| CE [18] | 11.2 | 26.9 | 34.8 | 25.3 | - | - | - | - |
| MAN (Non-Transformer) | **11.8** | **27.9** | **36.0** | **24** | - | - | - | - |
| MMT [6] | 13.2 | 29.2 | 38.8 | 21 | 12.1 | 29.3 | 37.9 | 22.5 |
| MAN (Transformer) | **13.6** | **30.0** | **39.4** | **20** | **12.5** | **29.8** | **38.7** | **22** |

Table 6: Retrieval performances on the LSMDC dataset.

**Results:** We report the evaluation results of MAN and the competing text-video retrieval methods on MSR-VTT (Table 5) and LSMDC (Table 6). In order to be fair, we still divide all the methods into two categories, Non-Transformer-based and Transformer-based methods. As for the former, MAN outperforms existing state-of-the-art methods in all the metrics on both MSR-VTT and LSMDC. Benefiting from the fine-grained temporal correlation of different modalities, the final representation contains not only the global multimodal information, but also local temporal interaction information. When we replace the backbone with Transformer, the metrics of MMT and MAN (Transformer) are very close, in other words, the effect of MAN is diminished. It may be because that the sample number of MSR-VTT and LSMDC is sufficient to stimulate the potential of Transformer.

## 4 Related Work

**Bilinear Model.** [5] proposes multimodal compact bilinear pooling network (MCB), which calculates the outer product between the visual and textual embeddings. The count-sketch projection is utilized to project the product on a lower-dimensional space. [15] proposes multimodal low-rank bilinear pooling network (MLB), where the full bilinear interaction between image and question spaces are parametrized by a tensor. To limit the number of free parameters, this tensor is constrained to be of low rank $r$. [1] proposes multimodal tucker network (MUTAN), which reduces the size of mono-modal embeddings while modeling their interaction as accurately as possible with a full bilinear integration scheme. [14] proposes bilinear attention network (BAN) to utilize the given vision-language information seamlessly by finding the bilinear attention distributions. Concretely, BAN considers bilinear interactions among two groups of input channels, while low-rank bilinear pooling extracts the joint representations for each pair of channels. Based on BAN, [2] and [17] propose relational reasoning by constructing a fine-grained graph between different input channels. [12] also tries to decompose the temporal projection tensor, but without theoretical support.

**Non-Linear Approximation.** Many endeavors have been devoted to the approximation of non-linear function. [13] employs a generalizable statistic expectation to approximate the non-linear kernels, specifically, [13] first defines a basic transformation (approximation of simply dot-product) and then utilizes Maclaurin expansion to expand out the non-linear functions to employ the conclusion of basic transformation. Such approximation accumulates two approximate errors. [26] proposes Tensor Sketching to approximate polynomial kernel functions based on Fourier transform. [5] develops pair-level integration of two vectors by leveraging the conclusions of [13] and [26], such method achieves

superior results on visual classification. Recently, [3] and [23] introduce the kernel decomposition into Transformer, they mainly approximate the exponential kernel with statistic expectation. The existing kernel decomposition methods are all proposed to process two vectors, considering the multimodal tasks, we first try to generalize the decomposition methods to more objects.

## 5 Conclusion

We propose a generalizable method called multi-linear attention network (MAN) to integrate as many modalities as possible. Compared with the existing multimodal joint representation learning methods, MAN conducts the hierarchical approximation decomposition (HAD) to reduce the exponential computational complexity. Considering that softmax function cannot be decomposed directly, we propose the addition random features mechanism (ARF) to reconstruct the non-linear operation with orthogonal projections. Furthermore, we employ local sequential constraints (LSC) to suppress the association scores of features that are virtually irrelevant. In the future, we would focus on how to combine powerful interaction methods (i.e. Transformer) with joint representation learning methods (i.e. MAN) more seamlessly.

## Acknowledgments

This work was supported in part by the National Key R&D Program of China under Grant No.2020YFC0832505, National Natural Science Foundation of China under Grant No.61836002, No.62072397 and Zhejiang Natural Science Foundation under Grant LR19F020006.

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
