# Generalized Multi-Linear Attention Network

**Tao Jin**
Zhejiang University
jint_zju@zju.edu.cn

**Zhou Zhao**[*]
Zhejiang University
zhaozhou@zju.edu.cn

## A  Appendix

## B  Detailed Process of MAN

The main process of MAN is shown in 1.

## C  Proof of Eq. 10 in the main paper

$$\sum_{t_1=1}^{T_1} ... \sum_{t_m=1}^{T_m} (\mathbf{1}^T(\prod_{j=1}^{m}\mathbf{B}_{j,t_j}))(\prod_{j=1}^{m}\mathbf{V}_{j,t_j}^T\mathbf{U}_j^{'}) = \mathbf{1}^T(\prod_{j=1}^{m}\mathbf{B}_j\mathbf{V}_j^T\mathbf{U}_j^{'}) \tag{1}$$

To evaluate the correctness of the above equation, we adopt the element-wise comparison. We utilize $\mathbf{X} \in \mathbb{R}^K$ and $\mathbf{Y} \in \mathbb{R}^K$ to denote the left and right formulas, respectively. Then, we calculate the $k$-th element of both $X$ and $Y$ as follows:

$$\begin{aligned}
\mathbf{X}_k &= \sum_{t_1=1}^{T_1} ... \sum_{t_m=1}^{T_m} (\mathbf{1}^T(\prod_{j=1}^{m}\mathbf{B}_{j,t_j}))(\prod_{j=1}^{m}\mathbf{V}_{j,t_j}^T\mathbf{U}_{j,k}^{'}) \\
&= \mathbf{1}^T \sum_{t_1=1}^{T_1} ... \sum_{t_m=1}^{T_m} \prod_{j=1}^{m}\mathbf{B}_{j,t_j}(\mathbf{V}_{j,t_j}^T\mathbf{U}_{j,k}^{'}) \\
\mathbf{Y}_k &= \mathbf{1}^T(\prod_{j=1}^{m}\mathbf{B}_j\mathbf{V}_j^T\mathbf{U}_{j,k}^{'})
\end{aligned} \tag{2}$$

To prove that $\mathbf{X}_k$ is equal to $\mathbf{Y}_k$, we evaluate the correctness of the following equation:

$$\sum_{t_1=1}^{T_1} ... \sum_{t_m=1}^{T_m} \prod_{j=1}^{m}\mathbf{B}_{j,t_j}(\mathbf{V}_{j,t_j}^T\mathbf{U}_{j,k}^{'}) = \prod_{j=1}^{m}\mathbf{B}_j\mathbf{V}_j^T\mathbf{U}_{j,k}^{'} \tag{3}$$

we still adopt element-wise comparison and utilize $\mathbf{X}^{'}, \mathbf{Y}^{'} \in \mathbb{R}^H$ to denote the left and right formulas, respectively. Then, we calculate the $h$-th element of $\mathbf{X}^{'}$ and $\mathbf{Y}^{'}$ as follows:

---

[*]Corresponding Author

35th Conference on Neural Information Processing Systems (NeurIPS 2021), Sydney, Australia.

---

**Algorithm 1** The detailed process of MAN.

---

1: **Input:** $\{\mathbf{V}_j \in \mathbb{R}^{d_j \times T_j} | j=[1,m]\}$, where $d_1=d_2=...=d_m=d$.
2: **Result:** MAN$(\{\mathbf{V}_j | j \in [1,m]\}; \mathbf{A}) \in \mathbb{R}^d$
3: Project each $\mathbf{V}_j \in \mathbb{R}^{d_j \times T_j}$ to multiple heads;
4: **for** $1...g$ **do**
5:     Project each $\mathbf{V}_j$ to $\mathbf{V}_j^T \mathbf{U}_j' \in \mathbb{R}^{T_j \times K}$ and $\mathbf{V}_j^T \mathbf{U}_j'' \in \mathbb{R}^{T_j \times K'}$, where $K=K'=d/g$;
6:     Generate the decomposed components of multi-linear attention map $\mathbf{A}$, $B_j \in \mathbb{R}^{H \times T_j}$, by leveraging Eq. 11 (main paper);
7:     Compute the denominator $D$ of softmax function with Eq. 12 (main paper);
8:     Compute the fusion result $\mathbf{f}' \in \mathbb{R}^K$ with Eq. 10 (main paper): $\mathbf{f}' = \mathbf{1}^T(\prod_{j=1}^m \mathbf{B}_j \mathbf{V}_j^T \mathbf{U}_j')$;
9:     Normalise $\mathbf{f}'$ with $D$ and employ pooling operation for it $\mathbf{f} = \mathbf{P}^T\mathbf{f}'$, where $\mathbf{f} \in \mathbb{R}^c$, $\mathbf{P} \in \mathbb{R}^{K \times c}$, $c=K$;
10: **end for**
11: Each head has the result $\mathbf{f}$, we impose some subscripts to distinguish them (i.e. $\mathbf{f}_1, ..., \mathbf{f}_g$). The final result is obtained by concatenating $\mathbf{f}_1, ..., \mathbf{f}_g$: $\mathbf{F} = [\mathbf{f}_1; ...; \mathbf{f}_g] \in \mathbb{R}^d$.

---

$$
\mathbf{X}_h' = \sum_{t_1=1}^{T_1} ... \sum_{t_m=1}^{T_m} \prod_{j=1}^m \mathbf{B}_{j,t_j,h}(\mathbf{V}_{j,t_j}^T \mathbf{U}_{j,k}')
$$

$$
\mathbf{Y}_h' = \prod_{j=1}^m \mathbf{B}_{j,:,h}(\mathbf{V}_j^T \mathbf{U}_{j,k}') = \prod_{j=1}^m \left[ \mathbf{B}_{j,1,h}(\mathbf{V}_{j,1}^T \mathbf{U}_{j,k}') + ... + \mathbf{B}_{j,T_j,h}(\mathbf{V}_{j,T_j}^T \mathbf{U}_{j,k}') \right] \tag{4}
$$

$$
= \prod_{j=1}^m \sum_{t_j=1}^{T_j} \mathbf{B}_{j,t_j,h}(\mathbf{V}_{j,t_j}^T \mathbf{U}_{j,k}')
$$

where $\mathbf{B}_{j,:,h} \in \mathbb{R}^{1 \times T_j}$ and $\mathbf{V}_j^T \mathbf{U}_{j,k}' \in \mathbb{R}^{T_j}$, $\mathbf{B}_{j,t_j,h}$ and $\mathbf{V}_{j,t_j}^T \mathbf{U}_{j,k}'$ are the $t_j$-th elements of them. Following the simple transformation like $(a+b)(c+d) = ac + ad + bc + bd$, we can obtain the fact that $\mathbf{X}_h'$ is equal to $\mathbf{Y}_h'$.

## D    Proof of Eq. 12 in the main paper

$$
\sum \mathbf{A} = \sum(\prod_{j=1}^m \mathbf{B}_j \cdot \mathbf{1}) \tag{5}
$$

We can rewrite the right formula:

$$
\sum(\prod_{j=1}^m \mathbf{B}_j \cdot \mathbf{1}) = \sum_{i=1}^h(\prod_{j=1}^m \mathbf{B}_{j,:,i} \cdot \mathbf{1}) = \sum_{i=1}^h(\prod_{j=1}^m \sum_{t_j=1}^{T_j}(\mathbf{B}_{j,t_j,i}))
$$

$$
= \sum_{i=1}^h(\sum_{t_1=1}^{T_1} ... \sum_{t_m=1}^{T_m} \prod_{j=1}^m(\mathbf{B}_{j,t_j,i})) \tag{6}
$$

$$
= \sum_{t_1=1}^{T_1} ... \sum_{t_m=1}^{T_m}(\sum_{i=1}^h \prod_{j=1}^m(\mathbf{B}_{j,t_j,i}))
$$

We then rewrite Eq. 9 in the main paper as:

$$
\mathbf{A}_{t_1,t_2,...,t_m} = \mathbf{1}^T(\prod_{j=1}^m \mathbf{B}_{j,t_j}) = \sum_{i=1}^h \prod_{j=1}^m(\mathbf{B}_{j,t_j,i}) \tag{7}
$$

since $\mathbf{A}_{t_1,t_2,\ldots,t_m}$ is an element of $\mathbf{A} \in \mathbb{R}^{\prod_{j=1}^{m} T_j}$, the equation is proved.

# E   Proof of Eq. 11 in the main paper

$$\mathrm{SM}(\{v_j | j \in [1, m]\}) = \exp(\sum_{j=1}^{m} \sum_{k=1, \neq j}^{m} v_k^T v_j)$$
$$= \exp\left(\|v_1 + \ldots + v_m\|^2/2\right) \cdot \exp\left(-\|v_1\|^2/2\right) \cdot \ldots \cdot \exp\left(-\|v_m\|^2/2\right) \tag{8}$$

Next, let $w \in \mathbb{R}^{K'}$. We use the fact that:

$$(2\pi)^{-K'/2} \int \exp\left(-\|w - c\|_2^2/2\right) dw = 1 \tag{9}$$

for any $c \in \mathbb{R}^{K'}$ and derive:

$$\exp(\|v_1 + \ldots + v_m\|^2/2) = (2\pi)^{-K'/2} \exp\left(\|v_1 + \ldots + v_m\|^2/2\right) \int \exp\left(-\|w - (v_1 + \ldots + v_m)\|^2/2\right) dw$$
$$= (2\pi)^{-K'/2} \int \exp\left(-\|w\|^2/2 + w^\top(v_1 + \ldots + v_m) - \|v_1 + \ldots + v_m\|^2/2 + \|v_1 + \ldots + v_m\|^2/2\right) dw$$
$$= (2\pi)^{-K'/2} \int \exp\left(-\|w\|^2/2 + w^\top(v_1 + \ldots + v_m)\right) dw$$
$$= (2\pi)^{-K'/2} \int \exp\left(-\|w\|^2/2\right) \cdot \exp\left(w^\top v_1\right) \cdot \ldots \cdot \exp\left(w^\top v_m\right) dw$$
$$= \mathbb{E}_{w \sim \mathcal{N}(0, \mathbf{I}_{K'})} \left[\exp\left(w^\top v_1\right) \cdot \ldots \cdot \exp\left(w^\top v_m\right)\right] \tag{10}$$

That completes the proof.

# F   Theoretical Error of Eq. 11 in the main paper

$$\mathrm{SM}(\{v_j | j \in [1, m]\}) = \exp\left(-\frac{\|\mathbf{v_1}\|^2 + \ldots + \|\mathbf{v_m}\|^2}{2}\right) \mathbb{E}_{w \sim \mathcal{N}(0, \mathbf{1}_{K'})} \left[\exp\left(w^\top z\right)\right] \tag{11}$$

where $z = v_1 + v_2 + \ldots + v_m$, based on the fact $\mathbb{E}_{w \sim \mathcal{N}(0, \mathbf{I}_{K'})} \left[\exp\left(w^\top z\right)\right] = \exp\left(\frac{\|z\|^2}{2}\right)$, then we can obtain:

$$\mathrm{MSE}\left(\mathrm{SM}_H(\{v_j | j \in [1, m]\})\right) = \frac{1}{H} \exp\left(-\left(\|\mathbf{v_1}\|^2 + \ldots + \|\mathbf{v_m}\|^2\right)\right) \mathrm{Var}\left(\exp\left(w^\top z\right)\right)$$
$$= \frac{1}{H} \exp\left(-\left(\|\mathbf{v_1}\|^2 + \ldots + \|\mathbf{v_m}\|^2\right)\right) \left(\mathbb{E}\left[\exp\left(2w^\top z\right)\right] - \left(\mathbb{E}\left[\exp\left(w^\top z\right)\right]\right)^2\right)$$
$$= \frac{1}{H} \exp\left(-\left(\|\mathbf{v_1}\|^2 + \ldots + \|\mathbf{v_m}\|^2\right)\right) \left(\exp\left(2\|z\|^2\right) - \exp\left(z^2\right)\right) \tag{12}$$
$$= \frac{1}{H} \exp\left(-\left(\|\mathbf{v_1}\|^2 + \ldots + \|\mathbf{v_m}\|^2\right)\right) \exp\left(\|z\|^2\right) \left(\exp\left(\|z\|^2\right) - 1\right)$$
$$= \frac{1}{H} \exp\left(\|z\|^2\right) \mathrm{SM}^2(\{v_j | j \in [1, m]\}) \left(1 - \exp\left(-\|z\|^2\right)\right)$$

where $H$ denotes the number of random features.

| Metric \Model | w/o CAO | w/o HAD | w/o LSC | MAN |
|---|---|---|---|---|
| Infer. Time (s) | $3.46 \times 10^{-1}$ | $3.52 \times 10^{-1}$ | $1.82 \times 10^{-1}$ | $1.92 \times 10^{-1}$ |

Table 1: Inference time of different variants on CMU-MOSI, where we set the batch size as 32 and record the inference time on whole test set with 689 samples with a RTX 3080Ti GPU (10 GB memory).

## G  Orthogonal Random Features (ORFs)

As mentioned in [1], to further reduce the variance of the estimator, we entangle different random weights $\boldsymbol{w}_1, \ldots, \boldsymbol{w}_H$ to be exactly orthogonal. This can be done while maintaining unbiasedness whenever isotropic distributions $\mathcal{N}(0, \mathbf{I}_{K'})$ are used by standard Gram-Schmidt renormalization procedure [2]. ORFs is a well-known method and can be applied to reduce the variance of softmax/Gaussian kernel estimators for any dimensionality $K'$ rather than just asymptotically for large enough $K'$ and leads to the first exponentially small bounds on large deviations probabilities that are strictly smaller than for non-orthogonal methods. One can refer to performer [1] for more details. The ORF mechanism requires $H \leq K'$, if $H > K'$, ORFs still can be used locally within each $K' \times K'$ block.

## H  More Experimental Details and Results

### H.1  Multimodal Sentiment Analysis and Speaker Traits Recognition

As for Non-Transformer-based MAN, we follow [5] to encode the multimodal features with LSTM first, and then the output is sent to the fusion module. [5] employs mean-pooling for the multimodal features along the temporal dimension and integrates multiple vectorial representations, leading to the global multimodal fusion. MAN could provide much local information for the original structure.

As for the Transformer-based MAN, we follow [6] to encode the multimodal features with Transformer first, and then the output is sent to the fusion module. Similarly, [6] only employs the global features of multiple modalities for the subsequent tasks, resulting in a loss of local dynamic information. In practice, we add MAN as an auxiliary module to predict a result that is integrated (simple addition operation) with the result of the original structure. The two fusion modules could provide complementary information for each other. All experiments are conducted on 4 RTX 3080Ti (10GB Memory).

### H.2  Multimodal Retrieval

The main idea of multimodal retrieval is similar to that of multimodal sentiment analysis. We refer to [4] and [3] for the implementation of backbones and keep the original structures, only adding MAN as an extra module. We further fuse their results for the subsequent tasks. Concretely, for MAN (Non-Transformer), we directly concatenate the representations of MAN and original multimodal fusion module and then match the final representation and text query. As for MAN (Transformer), we employ an extra branch to match the text representation and the output of MAN and then assign the corresponding weight (We assign weights to each module equally in practice). Since [4] and [3] only employ the global multimodal features for multimodal fusion. The fine-grained temporal information captured by MAN has an impact on the performances. All the experiments are conducted on 4 RTX 3080Ti (10GB Memory).

### H.3  About Inference Time

Since the inference time is greatly influenced by the implementation of the codes, we implement many versions for the model without HAD. The core idea is to reduce the number of loops in the implementation. Finally, we reduce the implementation loops from $T^3$ to $T$ on CMU-MOSI with pytorch matrix calculation. The results are shown as follows:

# I Drawbacks and Future Work

The core idea of this paper is to provide an approximate calculation method of multimodal integration from a theoretical point of view. However, when the number of samples is enough or the extracted features are powerful, the effectiveness of MAN will be greatly influenced. Since Transformer and Bert are the mainstream multimodal interaction methods currently, MAN lacks compatibility with them and the random features approximation is unstable to some extent. Therefore, How to be compatible with the powerful network is the problem to be solved in the future