# OpenReview forum: "Generalizable Multi-linear Attention Network"
_NeurIPS.cc/2021/Conference — NeurIPS 2021 Poster_

### Official Review · Reviewer_ERDD · 2021-06-29

**Rating:** 8
**Confidence:** 5

**Summary:**

For efficient multimodal learning having more than two modalities, they generalized the bilinear attention networks (Kim et al., 2018) to multi-linear attention networks using the same gist of low-rank pooling (Kim et al., 2017) and the sampling-based softmax approximation (Choromanski et al., 2021). To reduce the computational cost to build a multi-linear attention tensor, the proposed sampling-based approach was a novel idea using the well-established softmax approximation in the transformer-based architectures. Their extensive experiments on four datasets of three different tasks confirm that the proposed method consistently outperforms the comparative methods while giving the ablation study to assess the sub-modules one-by-one and the impact of the number of samples used in the softmax approximation.

**Limitations And Societal Impact:**

Please refer to the suggestion in the main review.

**Main Review:**

Overall, this work is well-written to deal with an interesting topic, more-than-two modality learning, in an efficient manner. The idea used in this approach is straightforward but having clever handling of the computational cost, which is one of the major concerns in practical multimodal learning. I believe the manuscript can be improved further by considering some points below:

### Clarity

c1) Please clarify the proposed method, "combinational addition operation (CAO) to replace the Hadamard product." (L109) I think it deserves to be more specifically elaborated.

c2) The $v_j$ in Eqn. 11 should be defined to prevent possible misleading.

c3) Could you compare with the other methods in terms of the number of parameters, FLOPS (as in Table 4), and inference time? Especially, the proposed method uses sampling to approximate the softmax operation. How much does it impact the computational time compared with the other methods or the controlled method which does not use the sampling?

c4) In the Performers, they said their random samples should be periodically redrawn for stable training (Choromanski et al., 2021). Doesn't it affect multimodal training?


### Suggestion

s1) The temporal order of Multimodal Temporal Features. The proposed one is not considering the temporal order. Why don't you use the two features, $(t_1, -1)$ and $(1, t_2)$, where the dot-product results in $t_1 - t_2$, to consider the temporal order instead of the absolute difference of $t_1$ and $t_2$ in L123?

s2) Could you add a paragraph dedicated to the methods using softmax approximation in Related Work?

s3) Please explicitly summarize the limitation of this work, which may include the time and space complexities as described in the comment of c3.


### Minors

m1) why do you call it "addition random features (ARF)"

m2) L71, rewrited -> rewritten

m3) L72,81, the rank is "at most" d, not the value of rank if U or V is not full rank.

m4) L104, "...employing softmax function may result in even attention distributions when the attention map is large." -> could you elaborate this sentence on what you want to mean by this?

----------
I willingly defend the paper and hold my score of 8. I do not see the idea as incremental from BAN or MUTAN. For the more-than-two modality learning, the computational cost of high-order tensor is prohibitive, aside from the theoretical application of previous bimodal methods. Because of that, I believe the novelty of this work from the combination of different but appropriate strategies to achieve the goal of *multimodal* learning, providing extensive empirical validations. However, it can be seen as having a limitation in a theoretical viewpoint, possibly due to relying on the previous core works.

**Time Spent Reviewing:**

5

---

> ### Author Response · Authors · 2021-08-10
> **Author response to Reviewer ERDD**
>
> Thanks for your thoughtful review. Below you will find our responses to your comments.
>
> **1.	Clarity of CAO**
>
> We illustrate the reasons for CAO in Lines 99-106.
>
> More specifically, we consider two ways to obtain the interaction logit $A_{t_1, t_2,..., t_m}$, one is to employ the Hadamard product of $m$ vectors $(\prod_{j=1}^{m} V_{j,t_j}^{T} U_{j}^{''}) \cdot \mathbf{1}$, the other (CAO) is to add up all the paired vectors $\sum_{j=1}^{m} \sum_{q=1,\neq j}^{m}  (V_{q,t_q}^{T} U_{q}^{''}) (U_j^{''T} V_{j,t_j})$ in Eq. 7. We carefully analyze the differences between two methods:
>
> - CAO could make the attention distribution more sparse. As the second reason in Lines 101-105, when the number of modalities is large, the size of interaction tensor $A$ will be exponentially big. Meanwhile, since the absolute values of most multiplicative factors are between 0 and 1, the Hadamard product will be small (i.e. $0.8^3 < 0.8^2 < 0.8$, the bigger the exponent, the smaller the elements). If we apply softmax for $A$, normalized attention weights of different groups would be close (i.e $\exp(0.5)$ vs $\exp(0.8)$). While Eq. 7 prevents the continual multiplication of elements (less than 1) and add up all the paired vectors, in this way, the attention weights of different groups would be more distinguishing (i.e. $\exp(0.5)$ vs $\exp(3)$).
> - CAO is well compatible with positional embeddings and LSC. As the third reason in Lines 105-107, existing positional constraints are mainly designed for the matrix multiplication of two vectors. The Hadamard product is not suitable .
> - Besides, CAO can be easily implemented borrowing the intuition of Performer, while Hadamard product may have theoretical difficulties.
>
> **2.	$v_j$ in Eq. 11**
>
> $v_j$ denotes the feature vector of $j$-th modality. We will revise the paper to prevent possible misleading.
>
> **3.	Parameter number and inference time of different variants**
>
> We supplement Table 4 with parameter number and inference time as follows:
>
> &#160;  &#160; &#160;  &#160; &#160;  &#160; &#160;  &#160; &#160;  &#160; &#160;  &#160; **Table: Some comparisons of one fusion layer**
>
> | Metrics/Model | w/o CAO | w/o HAD | w/o LSC | MAN |
> | ------ | ------ | ------ | ------ | ------ |
> | FLOPs | $8.31 \times 10^6$ | $10.23 \times 10^6$ | $3.07 \times 10^5$ | $3.87 \times 10^5$ |
> | Inference(s) | $3.46 \times 10^{-1}$ | $3.52 \times 10^{-1}$ | $1.82 \times 10^{-1}$ | $1.92 \times 10^{-1}$ |
> | Params(M) | $9.6 \times 10^{-3}$ | $9.6 \times 10^{-3}$ | $9.9 \times 10^{-3}$ | $10.2 \times 10^{-3}$ |
>
> We find that the inference time (we record the total time of all 686 test-samples) is related to FLOPs to some extent. Comparing with w/o HAD, MAN could achieve a large improvement. A small problem is the additional parameters of projection layers of random maps, but we argue that these parameters can be neglected considering the total parameters of the whole model.
>
> **4.	The impacts of redrawing**
>
> - The redrawing mechanism has no apparent effects on CMU-MOSI and POM. We analyze that the CMU-MOSI and POM are relatively small compared with other datasets. But we usually test multiple runs, different sampling results may lead to fluctuant metrics.
>
> - The redrawing mechanism has effects on MSR-VTT and LSMDC.
>
> &#160;  &#160; &#160;  &#160; &#160;  &#160; &#160; **Table: Text-to-Video Retrieval Performances**
>
> | Model/Metrics | R@1 $\uparrow$ | R@5 $\uparrow$ | R@10 $\uparrow$ | MdR $\downarrow$ |
> | ------ | ------ | ------ | ------ | ------ |
> | MAN (w/o Redraw, MSR-VTT) | 21.4 | 50.6 | 63.2 | 5 |
> | MAN (Redraw, MSR-VTT) | 21.4 | 51.6 | 63.9  | 5 |
> | MAN (w/o Redraw, LSMDC) | 11.5 | 27.4 | 35.2 | 24 |
> | MAN (Redraw, LSMDC) | 11.8  | 27.9 | 36.0 | 23 |
>
> **5.	The suggestion for LSC**
>
> Suppose the number of time steps is 10, we define the second time step (2, -1), the first time step (1, 1). Actually, we can obtain the result of first and second time steps, however, in this way, to define the third time step (x, y) which satisfies (1). 2x-y = 1, (2). x+y = 2, we obtain (1, 1), the result is same as the first time step. So the reviewer’s method may be not suitable for more than two time steps.
>
> **6.	The related work of approximating kernel functions**
>
> Many endeavors have been devoted to the approximation of non-linear function. [1] employs a generalizable statistic expectation to approximate the non-linear kernels, specifically, [1] first defines a basic transformation (approximation of simply dot-product) and then utilizes Maclaurin expansion to expand out the non-linear functions to employ the conclusion of basic transformation. Such approximation accumulates two approximate errors. [2] proposes Tensor Sketching to approximate polynomial kernel functions based on Fourier transform. [3] develops pair-level fusion of two vectors by leveraging the conclusions of [1] and [2], such method achieves superior results on visual classification. Recently, [4] and [5] introduce the kernel decomposition into Transformer, they mainly approximate the exponential kernel with statistic expectation. The existing kernel decomposition methods are all proposed to process two vectors, considering the multimodal tasks, we first try to generalize the decomposition methods to more objects.
>
> **7.	Future improvements**
>
> - As stated in the Conclusion Section, we will exploit better way to make MAN and Transformer compatible in the future.
>
> **8. The name of ARF**
>
> In Eq. 7, we add up all the paired results for implementing CAO, so we call it addition random features mechanism.
>
> **9. Minors 2-4**
>
> Firstly, we will revise the typos 2, 3. Secondly, for the minors 4, the illustration of even and sparse distributions of Hadamard product and CAO is mentioned in the first rebuttal section.
>
> **Reference:**
>
> [1]. Kar P, Karnick H. Random feature maps for dot product kernels[C]//Artificial intelligence and statistics. PMLR, 2012: 583-591.
>
> [2]. Pham N, Pagh R. Fast and scalable polynomial kernels via explicit feature maps[C]//Proceedings of the 19th ACM SIGKDD international conference on Knowledge discovery and data mining. 2013: 239-247.
>
> [3]. Gao Y, Beijbom O, Zhang N, et al. Compact bilinear pooling[C]//Proceedings of the IEEE conference on computer vision and pattern recognition. 2016: 317-326.
>
> [4]. Choromanski K, Likhosherstov V, Dohan D, et al. Rethinking attention with performers[J]. arXiv preprint arXiv:2009.14794, 2020.
>
> [5]. Peng H, Pappas N, Yogatama D, et al. Random Feature Attention[C]//International Conference on Learning Representations. 2020.

---

### Official Review · Reviewer_G3yM · 2021-07-14

**Rating:** 6
**Confidence:** 5

**Summary:**

This paper extends the bilinear attention network (BAN) to process more modalities. Through production operation, the proposed model can accept multiple modalities as input. Then the authors utilize CAO, LSC, HAD and ARF to optimize the multi-linear model. This model achieves some great results on several tasks. However, there still remain some problems.

**Limitations And Societal Impact:**

Yes

**Main Review:**

Pros:
1. In this paper, the approximation of softmax seems a little interesting.
2. In experiments, the proposed model can derive some good results.

Cons:
1. The fusing method is not really novel for the reason that the fundamental idea is BAN and utilizing a high-order tensor to mix the modalities has been proposed, like Mutan[1].
2. On the softmax approximation:
    (a) Although I can find some novelty in the softmax approximation, I do not understand why the authors have to decompose softmax(A) rather than A. The model compression and flop reduction come from decomposition on A. So what is the advantage of decomposing softmax(A)?
    (b) It can be better to add more details on the softmax approximation. Specifically, this approximation can not replace the exp operation, so what is the concrete effect of the approximation? And how this approximation can help with decomposition?
    (c) In the ablation experiment, this paper discussed the influence of the random feature numbers on performance. However, it is weird that the performance of MAN decreases at about number 33 in the left figure and about 29 in the right figure. This phenomenon can be a fatal instability of MAN. Can the authors explain the reason? In addition, as an approximation, the analysis of how the random feature numbers influence the similarity between ARF and w/o HAD is necessary.
    (d) Will the feature number influence the time consumption?
3. The proposed model does not improve appear on the Transformer comparison.

[1]Ben-Younes, Hedi, et al. "Mutan: Multimodal tucker fusion for visual question answering." Proceedings of the IEEE international conference on computer vision. 2017.

**Time Spent Reviewing:**

10

---

> ### Author Response · Authors · 2021-08-10
> **Author response to Reviewer G3yM**
>
> Thanks for your thoughtful review. Below you will find our responses to your comments.
>
> **1.	The novelty of MAN**
>
> BAN, MUTAN, and our proposed MAN are all tensor-based methods, but we argue MAN has several unique innovations which BAN and MUTAN do not have (such innovations are strongly supported by reviewer ERDD, one can refer to the summary of reviewer ERDD):
> - Only MAN with hierarchical decomposition can generalize to as many modalities as possible with linear complexity (considering both temporal and feature dimensions), BAN and MUTAN do not have such generalization. (**This is the original innovation of MAN**).
> - As highlighted by reviewer ERDD, the biggest innovation of MAN is the well-established sampling-based softmax approximation which can greatly reduce the computational complexity. **To the best of our knowledge, it is the first time to employ non-linear kernel decomposition for more than two vectors**. (BAN and MUTAN do not consider the non-linear function like softmax, thus our work has greater difficulty and contribution.). The details are illustrated in the following section.
> - The proposed local sequential constraints (LSC) module can well address the temporal positional relations of different multimodal sequential chunks. (**This is the original innovation of MAN**).
>
>
> **2.	Details of MAN**
>
> **2.1.	 Why decompose softmax(A), not A?**
>
> The approximation of $\text{softmax}(A)$ function is the biggest innovation of MAN. We guess that the reviewer may consider that “firstly decomposing $A$ and then applying softmax function for $A$” is easier than approximation of $\text{softmax}(A)$. However, the former operation is mathematically impossible. Taking an easy example, suppose we have two modalities $V_1 \in \mathbb{R}^{T_1 \times d_1}$ and $V_2 \in \mathbb{R}^{T_2 \times d_2}$, we employ multi-linear pooling to fuse each time-step group and obtain $V \in \mathbb{R}^{T_1 \times T_2 \times d}$, where $d$ is the output dimension of pooling operation. The attention weights for all time-step groups are $\text{softmax}(A) \in \mathbb{R}^{T_1 \times T_2}$. The tensor multiplication of $\text{softmax}(A) \cdot V$ results in a $d$-dimensional vector. To reduce the exponential complexity of time steps ($O(\prod_{j=1}^{m} T_j)$, $m$ is 2 in this example), we want to decompose the attention weights $\text{softmax}(A)$ into $B_1 \cdot B_2^T$, where $B_1 \in \mathbb{R}^{T_1 \times h}, B_2 \in \mathbb{R}^{T_2 \times h}$. Under the mathematical derivations (Eqs. 9, 10), $B_1$ and $B_2$ can operate directly with $V_1$ and $V_2$, leading to the complexity reduction ($O(\sum_{j=1}^{m} T_j)$). However, decomposing $A$ inside the softmax function cannot let the obtained $B_1$ and $B_2$ directly operate with $V_1$ and $V_2$, since the softmax function blocks them from meeting. In other words, if the softmax function is not approximated, the calculation with complexity $O(\prod_{j=1}^{m} T_j)$ of attention weights always exist.
>
> **2.2.	The approximation of exp function and the complexity reduction**
>
> The reviewer may misunderstand the process of MAN. We mainly approximate exp function for the purpose of approximating softmax function (they are almost equivalent, the only difference between exp and softmax is normalization, which is also easy to implement). The reviewer can refer to Eq. 11 and Line 140, we have illustrated that the approximation can simulate the non-linear exp operation. The detailed proof is in appendix Lines 18-21.
>
> For the illustration of complexity reduction, the reviewer can refer to Lines 163-166. Without the approximation decomposition, the calculation of attention weights need exponential complexity $O(\prod_{j=1}^{m} T_j)$, while employing decomposition, the complexity becomes linear $O(\sum_{j=1}^{m} T_j)$. The corresponding derivations are Eqs. 5, 9, 10 and detailed proof is shown in appendix Lines 4-13, which the reviewer can refer to. The most intuitive reason is that the $A$ in Eq. 5 (obtained with the major complexity) can be decomposed to multiple low-order tensors (multiple ${B}_{j,t_j}$ in Eq. 9). In other words, Eq. 9 directly reduces the complexity.
>
> **2.3.	The fluctuations of curve about number of random features**
>
> Since the approximation of $\text{softmax}(A)$ employs the expectation of randomly generated variables (the $w$ in Eq. 11), between the adjacent feature numbers, it is normal and possible that more random features may get bigger error (but it will be a small difference), results may have small negligible fluctuations between adjacent feature numbers due to the degree of approximation varies with each run. However, we mainly focus on the overall trend and find that the overall trend is steadily rising, further, we argue that the small fluctuations only exist in adjacent feature numbers and would not affect the overall trend of nonadjacent feature numbers. When the random features are enough (about $20$ random features) for accurate approximation, the metrics of MAN are going to be around those of w/o HAD, such results strongly demonstrate the success of approximation.
>
> **2.3.	The analysis of approximation error**
>
> The reviewer can refer to the appendix Lines 22-25. As for the analysis of approximation error, we give the detailed derivations of approximation error between ARF and w/o HAD in the appendix Lines 22-25, the bigger the number of random features, the smaller the error. Such theoretical analysis agrees well with our experimental results (the overall rising trend with small negligible fluctuations). In the curve Fig. 2, the results show that about $20$ random features are enough to accurate approximation.
>
> **2.4.	The time consumption brought by feature number**
>
> **Table: The total inference time of all 686 test-set samples in one fusion layer**
>
> | Metrics/Model | 5 (HAD) | 15 (HAD) | 24 (HAD) | w/o HAD |
> | ------ | ------ | ------ | ------ | ------ |
> | Inference(s) | $1.48 \times 10^{-1}$ | $1.74 \times 10^{-1}$ | $1.92 \times 10^{-1}$ | $3.52 \times 10^{-1}$ |
>
> We find that the feature number (24 is enough for approximation) has a slight effect on speed. However, hierarchical approximation decomposition (HAD) greatly reduces the running time, so the influences of feature number can be neglected.
>
> **3.	The improvement on Non-Transformer and Transformer backbones**
>
> We conduct both Transformer-based and Non-Transformer-based variants to evaluate the effectiveness of MAN and experimental results show the strong real-world values of MAN. For more details, the results on four datasets are as follows:
> - On POM, our proposed MAN (Transformer) could achieve 2.8375% improvement over Transformer-based SOTA metrics (average accuracy of 16 traits), while MAN (Non-Transformer) achieves 2.6125% improvement over Non-Transformer-based SOTA metrics, **even the improvement on Transformer is more significant**.
> - On CMU-MOSI, MAN surpasses all the baseline methods in all the metrics. The improvement of MAN (Transformer) is less than that of MAN (Non-Transformer) over the competitors. However, only MulT in the table reports the best metrics (upper bound) in the original paper, which is different from the fair evaluation rules of other methods (usually we average the results of ten experiments with random seeds). To keep the consistency, we directly copy the upper-bound unfair results in MulT paper, while the real metrics of MulT implemented by us are less. **MAN (Transformer) could achieve 1-2% apparent point improvement in all the metrics over MulT**.
> - On MSR-VTT and LMSDC, MAN also outperforms all the baseline methods in all the metrics. The improvement of MAN (Transformer) is a little bit less than that of MAN (Non-Transformer) over the corresponding competitors, **but it is still apparent**. As highlighted in our analysis in Line 292, it may be because that the sample number of MSR-VTT and LSMDC is sufficient to stimulate the potential of Transformer.
> - **Therefore, we argue that the improvement (Transformer) of all the datasets is apparent to some extent.**
>
> The supplementary real-world values of MAN:
>
> - Generally, adequate training samples are not always available in the real-world applications. With low-resource condition, Transformer-based and Non-Transformer-based methods have very close results (i.e. the results on POM dataset).The improvements of MAN (Transformer) and MAN (Non-Transformer) are close. When there are plenty of training samples, the effectiveness of MAN (Transformer) may be slightly diminished (MAN (Non-Transformer) is not affected), but it still brings apparent improvement to all the metrics. Therefore, we obtain the conclusion that MAN (Transformer) has good effects and real-world values for both low-resource and abundant-resource conditions, just the improvement on low-resource condition is a little bit more apparent, besides, MAN (Non-Transformer) has similar effects for both conditions.
> - Besides, online applications require lightweight structure with acceptable metrics, even the training samples are enough, Transformer may be not the best option. In this condition, MAN also can show its values.

---

### Official Review · Reviewer_62Vh · 2021-07-15

**Rating:** 7
**Confidence:** 3

**Summary:**

Motivated by the high computational cost of multi-modal multi-linear pooling, this paper proposed an efficient attention based multi-modal fusion mechanism. Low-rank bilinear pooling [1] is expended to multi-modal multi-linear case, following with the Multi-Linear Attention Network (MAN). The MAN combines cross modal attention and temporal feature, which is also decomposed based on Performers. Experiments show that the FLOPs decrease with a stable performance on multimodal down-stream task.

**Limitations And Societal Impact:**

The author mostly evaluated the strengths but rarely analyze the weaknesses of their work such as what if the length of the modality sequence is too short, in my opinion, shorter sequence length may influence the performance of the HAD part.

According to the list of potential negative societal impacts provided in Ethics Guidelines, there is no potential negative societal impact in this submission.

**Main Review:**

The proposed Multi-Linear Attention Network (MAN) gets noticeable experimental results on various of multimodal tasks, including Multimodal Sentiment Analysis, Speaker Traits Recognition and Video Retrieval. Both transformer-free backbone and transformer-based backbone can benefit from this generalizable method. However, it would be better if some detailed issues can be settled as follow:

In the Approach section, the decomposition of bilinear pooling and efficient multi-modal attention are widely explored in the past few years. This paper combined the well-known techniques [2, 3, 4, 5]. However, the task is novel to some extent, the author expands these methods to multi-modal case and combine them effectively. So, it would be better if the proposed MAN can be described more explicit. Specifically, how the multi-linear attention map works cross modalities and time chunks, or how the Hierarchical Approximation Decomposition decrease the FLOPs. What’s more, the differences between MAN and Performers should be explained more clearly for novelty. It’s not clear how this work differs from previous contributions, but the related works are adequately cited.

In the Related Work section, only some mathematical works are cited. However, the proposed MAN is proposed to applied in multi-modal tasks, so, it would be better if some multimodal works are cited. In particular, the audio-visual-text task--Video-Audio Scene-Aware Dialog, is mostly related to this work, so some related works can be included [7,8], similar with Vide Retrieval [9,10].

In the Experiments section, the proposed MAN gets great performance, and it may be better if some other experimental results are included. Firstly, the paper is focus on multi-modal multi-linear model, and the computational cost is related to not only the number of modality but also the length of the modality sequence. So, in my opinion, the experiments would be better if the discussion of sequence length is included as [2]. Similarly, the performance may not be promised when the sequence length is small. Secondly, in the Ablation Study subsection, only two tables are given to show the performance and the FLOPs, however, the discussion about the time cost is not describe clearly, so it would be better if this issue is added to the Ablation Study. Thirdly, the sequence attention gets great improvement in the submission, to get temporal context in a multimodal sequence, what are the advantages of your proposed method LSC compared with early temporal extraction, such as using temporal backbone [6, 7]. It may be more abundant if this point is discussed.

In general, the submission is clearly written with great experimental performance, the claims are well supported with enough theoretical analysis, but the novelty of it is limited. It would be more convincing if the Ablation Study is expended. The methods used is appropriate and efficient and the work is a complete work, which can be applied as an additional module to most multimodal fusion structure. And others are likely to use the ideas or build on them. The submission addresses a difficult task in a better way than previous work and advances the state of the art in a demonstrate way.

Post Rebuttal:

The authors' response has well addressed my previous concerns, especially the clarification of multi-linear attention map and hierarchical approximation decomposition and the differences between MAN and Performers. Moreover, the authors also accordingly provide some new experimental results w.r.t. my previous comments. Such clarification and results has further strengthened the core contribution of this paper. Hence, I decide to raise my score to 7.

**Time Spent Reviewing:**

10

---

> ### Author Response · Authors · 2021-08-10
> **Author response to Reviewer 62Vh**
>
> Thanks for your thoughtful review. Below you will find our responses to your comments. Besides, if we understand correctly, reviewer may forget to paste the references 1-10 in the comments, so we try our best to guess the intentions in some responses.
>
> **1. The Approach Section**
>
> **1.1.	The illustration of multi-linear attention map**
>
> For the illustration of multi-linear attention map, we take an easy example. Suppose we have two modalities $V_1 \in \mathbb{R}^{T_1 \times d_1}$ and $V_2 \in \mathbb{R}^{T_2 \times d_2}$, we could employ multi-linear pooling for each time-step group (i.e. the $t_1$-th time step of $V_1$ and the $t_2$-th time step of $V_2$), the total number of groups is $T_1 \times T_2$, so the final fusion result is $V \in \mathbb{R}^{T_1 \times T_2 \times d}$, where $d$ is the dimension of linear pooling operation. With the similar way like multi-linear pooling, we can obtain $A \in \mathbb{R}^{T_1 \times T_2}$, just the pooling dimension becomes 1, we then apply softmax function for $A$ to obtain the **multi-linear attention map** $\text{softmax}(A) \in \mathbb{R}^{T_1 \times T_2}$, each time-step group has its corresponding attention weight, the tensor multiplication is developed for $\text{softmax}(A) \cdot V$, the final result is a $d$-dimensional vector. The process is similar for more modalities.
>
> **1.2. The illustration of hierarchical approximation decomposition (HAD)**
>
> For the illustration of HAD, we give a simple illustration about computational complexity in Lines 163-166. With hierarchical approximation decomposition (HAD), the complexity becomes from original exponential pattern $O(\prod_{j=1}^{m} T_j)$ to acceptable linear pattern $O(\sum_{j=1}^{m} T_j)$, leading to the reduction of FLOPs simultaneously. As for the detailed theoretical analysis, the reviewer can refer to Eqs. 5, 9, 10 and the mathematical derivations in the appendix Lines 4-13, the most intuitive reason is that the large-size $A$ in Eq. 5 (obtained with the major complexity) can be decomposed to multiple low-order tensors (multiple ${B}_{j,t_j}$ in Eq. 9). In other words, Eq. 9 directly reduces the complexity.
>
> **1.3.	The differences between MAN and Performer [1] (the innovations of MAN)**
>
> We only borrow the intuition of random features approximation from Performer [1], but the usage of approximation for as many modalities as possible (Performer can only process two vectors) and the other contributions (HAD, LSC) in our paper are almost all original.
>
> We highlight three innovations at the end of Introduction (such innovations are strongly supported by reviewer ERDD).
>
> - MAN is the extension of BAN. To reduce the complexity, we devise hierarchical approximation decomposition mechanism for the interaction tensor (**This is the original innovation of MAN**).
>
> - We adopt the addition random features mechanism to approximate the softmax function with the interaction of as many modalities as possible, while Performer can only process two vectors (**we borrow the intuition from Performer and develop an innovative task-specific module for our task, such innovation is also supported by reviewer ERDD**).
>
> - We introduce the local sequential constraints to process multimodal sequential features (**This is the original innovation of MAN**).
>
> **2.  The Related Work**
>
> Thanks for the advice, due to the space limitation, we cite and discuss the works that are related to our theory. Following the advice of reviewer, we will cite and discuss these recommended works in the revised version. But the reviewer may forget to put the titles of references.
>
> **3.  The Experimental Section**
>
> **3.1	The discussion of sequence length**
>
> The reviewer may misunderstand the video data of our paper, the video clips used in this paper are all short-sequence, not long-sequence. The used datasets have their fixed protocols for fair comparisons, for example, we sample 20 time steps for CMU-MOSI and POM, 30 time steps for MSR-VTT and LSMDC. Therefore, the experimental results have demonstrated the effectiveness of MAN for short sequence.
>
> Besides, the results of similar paper [1], Performer, reveal that the length of sequential data only affects the computational cost (time cost), not influencing the object metrics, like accuracy. The computational cost of Performer with random feature mechanism will be reduced a lot when the sequence is long enough. However, absolutely different from Performer, the computation reduction of MAN comes from hierarchical approximation  decomposition (HAD), not additional random feature mechanism (ARF). Therefore, even the video clips that we used are short, the FLOPs of MAN is much less than that of w/o HAD ($3.87 \times 10^{5}$ vs $10.23 \times 10^{6}$).
>
> | Metrics/Model | w/o HAD | MAN |
> | ------ | ------ | ------ |
> | FLOPs (20 steps) | $10.23 \times 10^{6}$ | $3.87 \times 10^{5}$ |
> |FLOPs (5 steps)|$1.87 \times 10^5$|$9.85 \times 10^{4}$ |
>
> To make the results more convincing, we re-sample the videos of CMU-MOSI to 5 time steps (The three modalities are all with 5 time steps). The difference of FLOPs is still apparent.
>
>
> **3.2.	The inference time of different variants**
>
> &#160;  &#160; &#160;  &#160; &#160;  &#160; &#160;  &#160; &#160;  &#160; &#160;  &#160; **Table: Some comparisons of one fusion layer**
>
> | Metrics/Model | w/o CAO | w/o HAD | w/o LSC | MAN |
> | ------ | ------ | ------ | ------ | ------ |
> | FLOPs | $8.31 \times 10^6$ | $10.23 \times 10^6$ | $3.07 \times 10^5$ | $3.87 \times 10^5$ |
> | Inference(s) | $3.46 \times 10^{-1}$ | $3.52 \times 10^{-1}$ | $1.82 \times 10^{-1}$ | $1.92 \times 10^{-1}$ |
>
> We mainly compare w/o HAD and MAN, the former generates a high-order interaction tensor, while MAN employs approximation decomposition. The total inference time (all 686 test-set samples testing on one GTX 3080Ti) is related to the FLOPs, the inference time of MAN is much less than w/o HAD.
>
> **3.3.	The comparisons of LSC**
>
> The reviewer may forget to paste references, so we have difficulty in guessing the intention. The local sequential constraints (LSC) are not used for temporal modeling, but for the positional correlation constraints, to some extent, like positional embeddings in Transformer [2] and Bert [3]. We also try the positional embeddings in Transformer and Bert and find that LSC could achieve better results. Compared with the traditional positional embeddings, LSC more forcefully defines the relationship between different time steps (The further apart the two time steps are, the lower the correlation), while the calculation results of traditional positional embeddings are uncontrollable. Besides, LSC can be combined with ARF perfectly.
>
> | Model/Metrics | BA$\uparrow$ | F1$\uparrow$ | MAE$\downarrow$ | Corr$\uparrow$ | MA$\uparrow$ |
> | ------ | ------ | ------ | ------ | ------ | ------ |
> | Sinc&Cos [2] | 78.4 | 78.3 | 0.920 | 0.672 | 36.7 |
> | Trainable [3] | 78.7 | 78.6 | 0.932 | 0.678 | 36.0 |
> |MAN| 79.6 | 79.4 | 0.892 | 0.697 | 37.8 |
>
> **Reference:**
>
> [1]. Choromanski K, Likhosherstov V, Dohan D, et al. Rethinking attention with performers[J]. arXiv preprint arXiv:2009.14794, 2020.
>
> [2]. Vaswani A, Shazeer N, Parmar N, et al. Attention is all you need[C]//Advances in neural information processing systems. 2017: 5998-6008.
>
> [3]. Devlin J, Chang M W, Lee K, et al. Bert: Pre-training of deep bidirectional transformers for language understanding[J]. arXiv preprint arXiv:1810.04805, 2018.

---

### Decision · Program_Chairs · 2021-09-27

**Decision:**

Accept (Poster)

**Comment:**

The reviewers are very positive about the paper and is clearly an accept (6,7,8). Perhaps the nature of the multimodal tasks that are chosen are slightly contrived making it not quite worth of a Spotlight.